# iNP_ESM: Neuropeptide Identification Based on Evolutionary Scale Modeling and Unified Representation Embedding Features

**DOI:** 10.3390/ijms25137049

**Published:** 2024-06-27

**Authors:** Honghao Li, Liangzhen Jiang, Kaixiang Yang, Shulin Shang, Mingxin Li, Zhibin Lv

**Affiliations:** 1College of Biomedical Engineering, Sichuan University, Chengdu 610041, China; lihonghao27@163.com (H.L.); shangshulin120@163.com (S.S.); limingxin1999@foxmail.com (M.L.); 2College of Food and Biological Engineering, Chengdu University, Chengdu 610106, China; jiangliangzhen@cdu.edu.cn; 3Country Key Laboratory of Coarse Cereal Processing, Ministry of Agriculture and Rural Affairs, Chengdu 610106, China; 4College of Software Engineering, Sichuan University, Chengdu 610041, China; yangkaixiang08@163.com

**Keywords:** neuropeptide identification, machine learning, protein language model

## Abstract

Neuropeptides are biomolecules with crucial physiological functions. Accurate identification of neuropeptides is essential for understanding nervous system regulatory mechanisms. However, traditional analysis methods are expensive and laborious, and the development of effective machine learning models continues to be a subject of current research. Hence, in this research, we constructed an SVM-based machine learning neuropeptide predictor, iNP_ESM, by integrating protein language models Evolutionary Scale Modeling (ESM) and Unified Representation (UniRep) for the first time. Our model utilized feature fusion and feature selection strategies to improve prediction accuracy during optimization. In addition, we validated the effectiveness of the optimization strategy with UMAP (Uniform Manifold Approximation and Projection) visualization. iNP_ESM outperforms existing models on a variety of machine learning evaluation metrics, with an accuracy of up to 0.937 in cross-validation and 0.928 in independent testing, demonstrating optimal neuropeptide recognition capabilities. We anticipate improved neuropeptide data in the future, and we believe that the iNP_ESM model will have broader applications in the research and clinical treatment of neurological diseases.

## 1. Introduction

Neuropeptides, molecular entities resembling proteins yet smaller in scale, function within the nervous system as either neurotransmitters or neuromodulators. Synthesized and secreted by neurons, these molecules can modulate the functioning of other neurons and various target cells [1,2,3]. These substances play a critical role in numerous physiological processes, including governing the flexibility and responsiveness of neural circuits [4], influencing emotional states such as mood, anxiety, and depression [2], orchestrating tissue repair and regeneration [5], and contributing to the pathogenesis of certain neurological conditions such as autism spectrum disorder [6].

Neuropeptides are distinguished by their significant structural heterogeneity, which includes a wide array of molecular sizes and variants. Furthermore, the concentration of neuropeptides in vivo is exceedingly low, typically within the picomolar to nanomolar range, presenting technical challenges for their detection and quantification [7,8]. It is imperative to employ advanced analytical methodologies, such as mass spectrometry, nuclear magnetic resonance spectroscopy (NMR), and immunohistochemistry, to investigate neuropeptides. These methods necessitate substantial technical expertise and can be both resource-intensive and time-consuming [7,8,9]. So the development of robust, interpretable computational frameworks for predicting neuropeptide structures remains a focal point of ongoing research endeavors.

With the advent of neuropeptide repositories, the application of data-centric computational paradigms has commenced within the domain of neuropeptide prognostication. In 2019, a machine learning-based neuropeptide prediction model called NeuroPIpred was first reported by Piyush Agrawal et al. [10], designed to predict insect neuropeptides. Wang et al. collected a database called NeuroPep [11], followed by the development of many advanced neuropeptide machine learning algorithms. In 2020, Bin et al. developed a dual-layered stacked model called PredNeuroP [12] by combining nine feature descriptors with five different machine learning algorithms for feature optimization. In 2021, Hasan designed a predictor, NeuroPred-FRL [13], in which a feature representation learning approach based on a two-step feature selection method chooses from 11 different encodings and 6 classifiers. Jiang also developed a stacked model, NeuroPpred-Fuse [14], integrating various features and selection methods. In 2023, Wang et al. first utilized the latest dataset, NeuroPep2.0 [15], adopting the protein language model to extract features and a convolutional neural network to strengthen local features, developing the NeuroPred-PLM model [16]. Despite the optimistic outcomes manifested by these machine learning constructs, they are invariably plagued by a constellation of challenges, including, but not limited to, a constrained quantity of neuropeptide predictions, the incorporation of redundant features, and a notable deficit in model interpretability [10,13,14,16,17,18].

Owing to the advancements in natural language processing (NLP) and deep learning (DL), protein language models (PLM) have rapidly emerged, which consider individual amino acids and their polymers as tokens, like words in the language [19]. It is adapted to derive fixed-sized vectors, called embeddings, for representing a given protein. From a theoretical standpoint, protein language models offer distinct advantages over traditional bioinformatics approaches [20]. Firstly, it obviates the need for the laborious steps of querying, alignment, and training by employing a singular forward-pass learning mechanism, significantly expediting the feature extraction process. Secondly, it leverages the vast diversity of protein corpora, comprising billions of sequences, to foster generalizability and optimize performance in downstream predictive tasks [21,22,23,24,25,26,27]. By adopting pre-training methods from natural language processing, these models can understand the properties and behaviors of proteins without relying on explicit structural data, achieving remarkable results in protein structure prediction, functional annotation, and the prediction of protein-ligand reactions [28,29,30,31,32,33,34,35,36].

In the current study, we have developed a new machine learning methodology, iNP_ESM, for the identification of neuropeptides. The feature extraction was performed with the use of protein language models, namely, Evolutionary Scale Modeling (ESM) and Unified Representation (UniRep) [21,25]. The neuropeptide identification algorithm implements the Support Vector Machine (SVM) algorithm. We subsequently optimized the model through feature fusion and selection techniques, and the optimized model was named iNP_ESM. Compared to the existing state-of-the-art NeuroPred-PLM, iNP_ESM demonstrated significant improvements in 10-fold cross-validation metrics, achieving an accuracy (ACC) of up to 0.937, a Matthews correlation coefficient (MCC) of up to 0.873, and an F1 score of up to 0.934. In independent test metrics, iNP_ESM achieved an ACC of up to 0.928, an MCC of up to 0.856, and an F1 score of up to 0.928. We hope our work will fill this gap and advance the development of machine learning models for neuropeptide prediction.

## 2. Results

The modeling process is illustrated in Figure 1. Initially, we input the amino acid sequences of neuropeptides into the pre-trained protein language models, obtaining a 1280-dimensional feature from ESM and a 1900-dimensional feature from UniRep. Subsequently, these features undergo fusion and selection processes before being input into an SVM machine learning model. Finally, after a comparative analysis of performance metrics, an optimized model is developed, which is designated as iNP_ESM.

### 2.1. Identifying the Optimal Baseline Models

To identify the optimal feature encoding method and machine learning algorithm, we combined six different features extracted from neuropeptide sequences with seven different machine learning algorithms, resulting in 42 baseline models. The scores of 10-fold cross-validation for each encoding under different algorithms are presented in Figure 2. It is evident that, except for the slightly lower scores of UniRep with GNB, the ESM and UniRep features consistently outperformed other features across various machine learning algorithms. Table 1 compares the accuracy of these baseline models based on 10-fold cross-validation. It was found that within each set of machine learning algorithms, the top-ranking feature was always ESM or UniRep. Specifically, UniRep achieved the highest scores with KNN and RF, at 0.895 and 0.908, respectively, while ESM achieved the highest scores with SVM, GNB, LDA, LGBM, and LR, at 0.764, 0.906, 0.920, 0.916, and 0.924, respectively. Notably, for both ESM and UniRep, the algorithm that consistently achieved the best performance was LGBM and SVM, with the best ACC scores for ESM being 0.920 and 0.924 under LGBM and SVM, respectively. This not only highlights ESM and UniRep.

As the best encoding methods for extracting neuropeptide features, but also indicate that LGBM and SVM are the most suitable machine learning algorithms for these features. Appendix A supplements this by comparing the independent test accuracy of the baseline models. Here, the UniRep feature achieved the highest score with KNN and RF at 0.885 and 0.896, while the ESM feature attained the highest scores across the other six algorithms: 0.777, 0.898, 0.914, 0.902, 0.892, and 0.918. These results demonstrate the superiority of ESM and UniRep in extracting neuropeptide features. Furthermore, we performed parameter optimization for the LGBM and SVM algorithms, with the results shown in Appendix A.

### 2.2. Feature Fusion Optimization

In general, integrating multiple features can capture more comprehensive information from a single dataset, thereby enhancing the robustness of the model. Based on the above-stated baseline models, we fused the features obtained from the 1280D ESM and the 1900D UniRep to get a new 3180D feature. We then applied the LGBM and SVM algorithms to this new feature, creating the new model ESM+UniRep_F3180. The results indicated that the models achieved better performance metrics under both algorithms. Figure 3 compares the cross-validation and independent test average scores of the ESM+UniRep_F3180 models with the optimized single-feature models. Appendix A provide supplementary details of the 10-fold cross-validation and independent test scores for the ESM+UniRep_F3180 and single-feature models. Comparing the average values, the LGBM model using the fused feature showed improvements in ACC scores by 0.9% to 1.2%, MCC by 1.8% to 2.4%, Sn by 0.2% to 0.9%, Sp by 1.5% to 1.6%, Pre by 1.4%, F1 by 0.8% to 1.2%, and AUC by 0.3% to 0.6%. For the SVM model with the fused feature, ACC improved by 0.5% to 0.7%, MCC by 1.0% to 1.4%, Sn by 0.9% to 1.4%, and AUC by 0.4%, while Sp did not improve, Pre increased by 0.1%, and F1 by 0.5% to 0.8%. These results demonstrate that the strategy of fusing ESM and UniRep effectively enhances the model’s ability to learn neuropeptide features. Additionally, the ESM+UniRep_F3180 model performed better with the SVM algorithm than with LGBM. Except for Sn, the average values for ACC, MCC, Sp, Pre, and F1 were higher. This suggests that SVM is a more suitable machine learning algorithm for the ESM+UniRep feature. Therefore, we selected SVM for constructing models in subsequent feature calculations.

### 2.3. Feature Selection Optimization and Visualization Analysis

Due to the large dimensionality of the ESM+UniRep_F3180, we applied feature selection strategies, successfully improving the model’s cross-validation metrics. Initially, we input the features of ESM+UniRep_F3180 into the LGBM algorithm, leveraging its built-in function to compute and rank the importance of each feature. Subsequently, we conducted a stepwise feature selection with a size of 5, selecting the top 5, 10, 15, 25, and so on, up to 300 features, and modeled them using SVM. It is important to note that SVM was chosen because, as previously observed, it performed better than LGBM with the fused feature. The changes in ACC and MCC scores during cross-validation and independent testing at each optimization step are shown in Figure 4. Firstly, as the number of features increased, the average ACC and average MCC rose quickly. When the number of features reached 120, both the average ACC and average MCC peaked at 0.929 and 0.859, respectively. Beyond this point, metrics showed a declining trend. Therefore, we selected the top 120 features to construct a new model named ESM+UniRep_F120 using SVM. Table 2 compares the metrics of the SVM-based models after feature fusion and selection. Compared to single-feature models, the ESM+UniRep_F120 model showed improved cross-validation metrics (ACC increased by 0.5~0.8%, MCC by 0.9~1.7%), validating the effectiveness of our optimization strategy. Additionally, comparing the two fused-feature models, we found that ESM+UniRep_F120 achieved higher cross-validation scores (ACC and MCC increased by 0.3% and 0.6%, respectively), while its independent test scores decreased (ACC and MCC decreased by 0.6% and 1.1%, respectively). We suppose this could be due to the reduction of redundant features, which enhanced learning on the training set and improved model robustness, but slightly reduced adaptability to entirely new data due to the loss of a significant number of features.

Additionally, we used UMAP to perform dimensionality reduction and visualization on the training sets of different features. Comparing Figure 5A,B with Figure 5C,D confirms that the clustering effect of fused features is superior to that of individual ESM and UniRep features. Comparing Figure 5C,D demonstrates that the 120D fused feature has comparable neuropeptide recognition capability to the 3180D fused feature but with significantly fewer features. These findings corroborate our previous data analysis results. We ultimately named the optimized model ESM+UniRep_F3180 as iNP_ESM_F3180 and the model ESM+UniRep_F120 as iNP_ESM_F120.

### 2.4. Comparison with Existing Methods

We also conducted a more in-depth performance comparison of our final model by comparing the iNP_ESM model with several existing methods. For fairness, all methods used the same latest training data, NeuroPep2.0, as iNP_ESM. The data source is NeuroPred-PLM, the first neuropeptide prediction model using NeuroPep2.0 data. Table 3 compares the independent test metrics of existing models. Our iNP_ESM_F3180 achieved the highest ACC (0.928) and MCC (0.856). The iNP_ESM_F120’s ACC (0.922) and MCC (0.845) were comparable to the current state-of-the-art model, NeuroPred-PLM. While the recall of iNP_ESM (0.930 and 0.917) slightly trailed that of NeuroPred-PLM (Rec = 0.941), iNP_ESM demonstrated superior results in precision (0.926 and 0.966) and F1 scores (0.928 and 0.927), surpassing NeuroPred-PLM (Pre = 0.907, F1 = 0.924). Further, Appendix A compares the 10-fold cross-validation metrics between iNP_ESM and NeuroPred-PLM, illustrating that iNP_ESM consistently outstrips NeuroPred-PLM across various metrics. Notably, iNP_ESM_F120 exhibited enhancements over NeuroPred-PLM in ACC, MCC, Sn, Sp, Pre, and F1 by 1.0%, 1.9%, 1.2%, 0.7%, 5.4%, and 0.7%, respectively. These findings collectively underscore the robustness of iNP_ESM as a formidable tool for neuropeptide identification.

## 3. Discussion

The development of the iNP_ESM model has significantly contributed to the field of neuropeptide identification through machine learning. By integrating ESM and UniRep protein language models with an SVM algorithm, we achieved superior performance metrics compared to existing models. Our results demonstrated marked improvements in evaluation metrics in both cross-validation and independent testing.

The success of iNP_ESM can be attributed to several key factors. Firstly, the fusion of ESM and UniRep features provided a comprehensive representation of neuropeptide sequences, capturing a broader range of biological properties and interactions. This fusion enhanced the model’s ability to distinguish neuropeptides from non-neuropeptides more accurately. Secondly, our feature selection process allowed us to optimize the model by retaining the most informative features while reducing redundancy, particularly when reducing the feature to 120 dimensions. Our study also highlighted the importance of visualization in model optimization. The use of UMAP for dimensionality reduction and visualization confirmed the superior clustering of neuropeptide features in the fused models, further enhancing interpretability—a common challenge in machine learning models.

Compared to NeuroPred-PLM, which utilizes only ESM representations, our model integrates a more diverse set of features (UniRep) while employing a reduced dimensionality (120D). Unlike complex stacked models such as NeuroPred-FRL, we implemented a simpler feature fusion and selection strategy, streamlining the modeling process and achieving superior results. However, the use of simplified features in our model can decrease accuracy in independent tests (Table 3). Achieving a balance between feature reduction and predictive efficiency remains an area for further exploration.

Despite these advancements, several challenges remain. The reliance on high-quality, comprehensive datasets is paramount for training robust models [37,38,39,40,41]. While the NeuroPep 2.0 dataset significantly improved our training process, the availability of more extensive and diverse neuropeptide data would further enhance model accuracy and generalizability. Additionally, there is still a need for more transparent methodologies that can provide deeper insights into the biological significance of the predictive features. Future work should focus on expanding the dataset, incorporating new neuropeptide sequences, and exploring other advanced protein language models to further refine feature extraction.

## 4. Materials and Methods

### 4.1. Dataset

Gathering datasets is a crucial step in machine learning, and we have utilized the latest NeuroPep 2.0 dataset [15]. NeuroPep 2.0 contains 11,417 unique neuropeptide records, nearly double that of its initial version. Neuropeptides included in the database must be verified through scientific literature or database annotations. These records are derived from 544 articles and 3688 precursor proteins from the UniProtKB database, covering 924 species and distributed across 81 neuropeptide families. In terms of data processing techniques, our methodology aligns with those employed by contemporary leading-edge algorithms such as NeuroPred-PLM. Initially, we exclude neuropeptides that do not conform to the specified length criteria of 5 to 100 amino acids. Subsequently, to avoid model overfitting, the remaining dataset was processed using CD-HIT with a 0.9 threshold to remove sequences with more than 90% similarity to other sequences [42]. Following these filtration processes, a total of 4463 neuropeptides were retained. For the purpose of an independent test, all selected neuropeptides were derived from the additions made in the NeuroPep 2.0 dataset, with 444 neuropeptides, constituting 10% of the total, chosen at random for this subset. Owing to the absence of experimentally verified negative instances, sequences were sourced from UniProt whose length distribution is similar to the positive examples [43], resulting in 4463 negative examples. From this collection, 444 sequences were randomly allocated to compose the non-neuropeptide control group for the test.

### 4.2. Feature Extraction Methods

In this paper, five encoding methods were utilized to represent neuropeptides for comparison. These methods include ESM, UniRep, TAPE_BERT [23], LM, BiLSTM, and Soft Symmetric Alignment Embedding (SSA) [44]. The training was conducted utilizing a 16-GB NVIDIA GPU.

#### 4.2.1. ESM

The training of ESM involved unsupervised learning on billions of protein sequences in UniRef50, enabling the model to capture a wide range of biological properties [25,45] The model produces 1280-dimensional output vectors. ESM makes use of a Masked Language Modeling Objective, in which a fraction of amino acids from the protein sequence are randomly masked while the model is forced to predict the masked amino acids. This task can be represented by the formula:(1)$$L_{\mathrm{MLM}}=-\underset{i\in M}{\sum }logp\left(x_{i}|x_{\backslash M}\right)$$

$x_{i}$ represents the masked amino acids, $x_{\backslash M}$ denotes the sequence excluding the masked positions, and M is the set of positions of the masked amino acids.

A key component of ESM is the use of the Transformer architecture. The Transformer model is designed with the concept of multiple encoders with stacked layers, each comprising a self-attention mechanism and a feedforward neural network. This approach was chosen because the Transformer architecture can structurally capture long-range dependencies and complex relational patterns in the data, which is crucial for understanding the intricate interactions that govern protein function and structure [46]. The self-attention mechanism, which is central to the Transformer model, enables the model to capture complex relationships between various positions. The key to the self-attention mechanism can be simplified by the following formula:(2)$$\mathrm{Attention}\left(Q,K,V\right)=\mathrm{soft}\max\left(\frac{QK^{T}}{\sqrt{d_{k}}}\right)V$$

Here, Q, K, and V represent the query, key, and value matrices. $d_{k}$ represents the dimension of the key vectors. This mechanism allows the model to capture global dependencies at every position in the sequence. The ESM model parameters used in this experiment can be obtained from the referenced literature [25].

#### 4.2.2. UniRep

UniRep has been trained using millions of unlabeled protein sequences from the UniRef50 [21]. The architecture of the UniRep model is the mLSTM recurrent neural network. One of the aims of UniRep training is to increase the prediction accuracy of the next amino acid, leading to decreased cross-entropy loss. This predictive task compels the UniRep model to incorporate representations of “cell state” or “hidden state”. For any specific protein sequence, the UniRep model’s output comprehensively captures various dimensions of the sequence’s structural and functional attributes, thereby providing a holistic characterization of the protein. The core of the mLSTM network is defined by the following equations [29]:(3)$$m_{t}=\left(X_{t}W_{xm}\right)⊗\left(h_{t-1}W_{hm}\right)$$
(4)$$f_{t}=\sigma \left(X_{t}W_{xf}+m_{t}W_{mf}\right)$$
(5)$$i_{t}=\sigma \left(X_{t}W_{xi}+m_{t}W_{mi}\right)$$
(6)$$o_{t}=\sigma \left(X_{t}W_{xo}+m_{t}W_{mo}\right)$$
(7)$$C_{t}=f_{t}⊗C_{t-1}+i_{t}⊗\tilde{C_{t}}$$
(8)$$\tilde{C_{t}}=\tanh\left(X_{t}W_{xf}+m_{t}W_{mh}\right)$$
(9)$$h_{t}=o_{t}⊗\tanh\left(C_{t}\right)$$

Here, ⊗ represents the element-wise multiplication of matrices. $m_{t}$ is the current multiplication interim state. $X_{t}$ is the input at the current timestep. W is the weight matrix. $h_{t-1}$ is the hidden state from the last time step. σ is the sigma activation function. $f_{t}$ is the output of the forget gate, which represents the previous memory retention. $i_{t}$ is the activation value of the input gate. $C_{t}$ is the cell state at the current time step, updated with the prior cell state $C_{t-1}$ and new candidate cell state $\tilde{C_{t}}$. $h_{t}$ is the current hidden state. $o_{t}$ is the activation of the output gate. The UniRep model parameters can be obtained from the referenced literature [21].

#### 4.2.3. TAPE_BERT

To evaluate the performance of various extraction methodologies, features were also encoded using TAPE_BERT, SSA, LM, and BiLSTM. TAPE (Tasks Assessing Protein Embeddings) is a benchmark testing framework [23]. TAPE utilizes the Pfam database [47] and employs architectures such as Transformer, LSTM, and ResNet to perform large-scale pre-training on protein sequences through self-supervised learning methods. Experimental results demonstrate that this self-supervised pre-training significantly enhances model performance in tasks such as protein structure prediction, remote homology detection, fluorescence prediction, and stability prediction. TAPE_BERT is based on the BERT architecture, and the formulas describing its principles are the same as those used to describe the Transformer in the previous introduction of ESM. The code is available in the paper [23].

#### 4.2.4. SSA, LM, and BiLSTM

SSA (Soft Symmetric Alignment) is a framework for learning protein sequence embeddings [44]. SSA is trained on the Pfam database [47] and uses BiLSTM encoders along with a soft symmetric alignment mechanism to learn structural information from protein sequences. Experiments show that the soft symmetric alignment mechanism significantly improves the accuracy of structure similarity predictions and performs well in tasks such as transmembrane region prediction. The formula for calculating the similarity score is as follows [44]:(10)$$\hat{s}=-\frac{1}{A}\overset{n}{\underset{i=1}{\sum }}\overset{m}{\underset{j=1}{\sum }}a_{ij}|z_{i}-z_{j}^{\prime }{|}_{1}$$
where $z_{i}$ and $z_{j}^{\prime }$ are the embedding vectors of two protein sequences, $a_{ij}$ is an element of the alignment matrix representing the alignment weight between positions i and j of the sequences, and A is the total length of the alignment:(11)$$a_{ij}=\alpha_{ij}+\beta_{ij}-\alpha_{ij}\beta_{ij}$$
(12)$$\alpha_{ij}=\frac{\exp\left(-{\left|z_{i}-z_{j}^{\prime }\right|}_{1}\right)}{\sum_{k=1}^{m}\exp\left(-{\left|z_{i}-z_{k}^{\prime }\right|}_{1}\right)}$$
(13)$$\beta_{ij}=\frac{\exp\left(-{\left|z_{i}-z_{j}^{\prime }\right|}_{1}\right)}{\sum_{k=1}^{n}\exp\left(-{\left|z_{k}-z_{j}^{\prime }\right|}_{1}\right)}$$
(14)$$A=\overset{n}{\underset{i=1}{\sum }}\overset{m}{\underset{j=1}{\sum }}a_{ij}$$

Additionally, we also extracted features LM and BiLSTM from the architecture of SSA embedding for comparison. These codes are available in the paper [44].

### 4.3. Machine Learning Methods

In this study, we employed seven widely used machine learning algorithms, specifically: K-Nearest Neighbors (KNN) [48], Support Vector Machines (SVM) [49,50], Logistic Regression (LR) [51], Random Forest (RF) [52], LightGBM (LGBM) [53], Linear Discriminant Analysis (LDA) [54], and Gaussian Naive Bayes (GNB) [55].

### 4.4. Feature Selection Methods

Feature selection is crucial for effectively constructing learning models through dimensionality reduction, removal of irrelevant or redundant attributes, reduction in training time, and augmentation of model performance and interpretability [56,57,58,59]. Typical examples of feature selection methods include Lasso and Ridge Regression [60], Random Forests [61], and Gradient Boosting Machines [62]. In previous studies on polypeptide sequences conducted in our laboratory [29], we tested various feature selection methods, including LGBM, ANOVA, and Chi-Squared. Ultimately, we chose the LGBM model due to its efficiency, simplicity, and ability to provide clear ranking results [63]. LGBM provides built-in functions to compute split importance. Specifically, during the construction of decision trees, LGBM records the features used for each node split. It calculates an importance score for each feature by adding the total number of times that feature is selected as a splitting node. Next, we sort the important features in descending order and choose a threshold to filter the important features.

### 4.5. Feature Visualization Methods

Data dimensionality reduction is a common data preprocessing technique that reduces the number of features in the data while keeping most of the vital information intact [64]. In machine learning, dimensionality reduction techniques are frequently coupled with data visualization to facilitate the understanding and interpretation of data and results. Notable dimensionality reduction visualization techniques include PCA (Principal Component Analysis) [65], t-SNE (t-distributed Stochastic Neighbor Embedding) [66], and UMAP (Uniform Manifold Approximation and Projection) [67,68]. UMAP visualizes the reduction by creating a graph of neighborhoods in a high-dimensional space and an optimization algorithm for the recreation of this graph in a lower-dimensional space. Compared to traditional methods, UMAP more effectively handles nonlinear data distributions, offers greater scalability, and operates at higher speeds. In this work, we utilized UMAP for dimensionality reduction and visualization.

### 4.6. Performance Evaluation Metrics

In this research, seven performance evaluation metrics have been adopted, including Accuracy (ACC), Matthews Correlation Coefficient (MCC), Sensitivity (Sn), Specificity (Sp), Precision (Pre), Recall (Rec), the F1 score, and the Area Under the Curve (AUC) [69,70]. The formulas are shown below:(15)$$\mathrm{ACC}=\frac{\mathrm{TP}+\mathrm{TN}}{\mathrm{TP}+\mathrm{TN}+\mathrm{FP}+\mathrm{FN}}$$
(16)$$\mathrm{MCC}=\frac{\mathrm{TP}\times \mathrm{TN}-\mathrm{FP}\times \mathrm{FN}}{\sqrt{\left(\mathrm{TP}+\mathrm{FP}\right)\times \left(\mathrm{TP}+\mathrm{FN}\right)\times \left(\mathrm{TN}+\mathrm{FP}\right)\times \left(\mathrm{TN}+\mathrm{FN}\right)}}$$
(17)$$\mathrm{Sn}=\frac{\mathrm{TP}}{\mathrm{TP}+\mathrm{FN}}$$
(18)$$\mathrm{Sp}=\frac{\mathrm{TN}}{\mathrm{TN}+\mathrm{FP}}$$
(19)$$\mathrm{Pre}=\frac{\mathrm{TP}}{\mathrm{TP}+\mathrm{FP}}$$
(20)$$\mathrm{Rec}=\frac{\mathrm{TP}}{\mathrm{TP}+\mathrm{FN}}$$
(21)$$F1=2\frac{\mathrm{Pre}\times \mathrm{Rec}}{\mathrm{Pre}+\mathrm{Rec}}$$

These metrics utilize four results from the classifier: True Positives (TP), True Negatives (TN), False Positives (FP), and False Negatives (FN).

## 5. Conclusions

In the essay, we created a novel neural peptide predictor named iNP_ESM. To construct the model, we compared various feature extraction methods and machine learning algorithms, ultimately selecting the protein language models ESM and UniRep. Following a series of optimizations, we obtained two optimal models based on SVM. Employing the same dataset, our iNP_ESM demonstrated state-of-the-art performance in cross-validation and independent testing. Visualization with UMAP highlighted that the strategies for feature fusion and selection significantly enhanced the model’s capability to extract features.

Despite the enhancements afforded by the protein language model in processing complex relationships within neural peptide prediction models, several issues and challenges remain. These include a lack of high-quality, large-scale datasets and the interpretability of deep learning models. As data resources expand and our understanding of protein language models deepens, we anticipate further improvements in the quality of neural peptide prediction models.

## Figures and Tables

**Figure 1 ijms-25-07049-f001:**
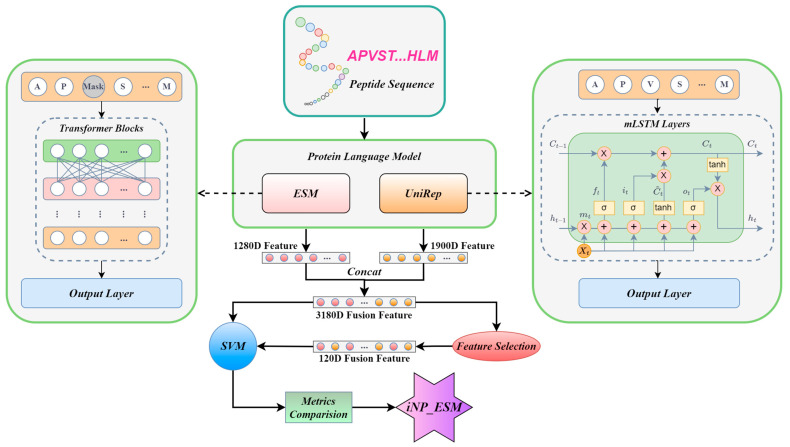
An overview of the iNP_ESM model. Initially, neuropeptide sequences are input into the protein language models ESM and UniRep, generating 1280D ESM features and 1900D UniRep features for each sequence. Subsequently, these features are combined to form a 3180D fused feature. This fused feature can be directly input into an SVM model. Alternatively, after dimensionality reduction through feature selection to 120 dimensions, the reduced feature can also be input into the SVM model. Following a series of optimizations and performance comparisons, the iNP_ESM model is finalized.

**Figure 2 ijms-25-07049-f002:**
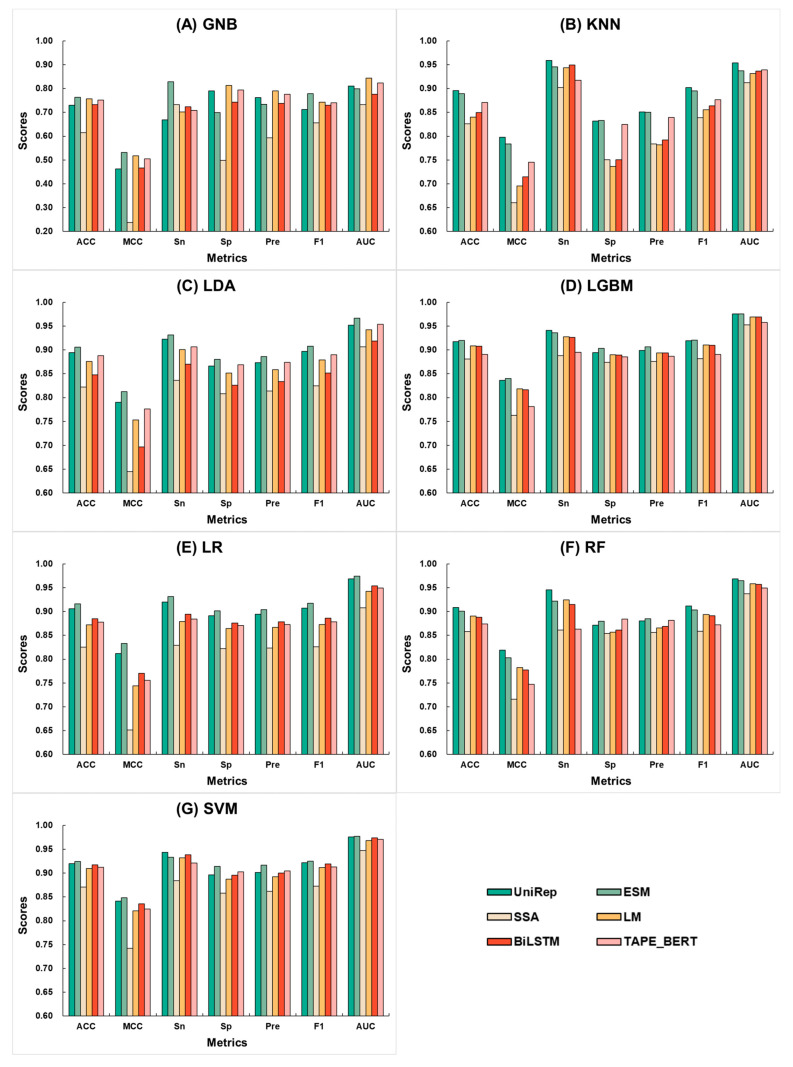
Comparison of 10-fold cross-validation metrics for the combination of six feature encoding methods and seven machine learning algorithms. Here, UniRep is represented in dark green, ESM in light green, SSA in light yellow, LM in dark yellow, BiLSTM in dark red, and TAPE_BERT in light red. The machine learning algorithms include (**A**) GNB, (**B**) KNN, (**C**) LDA, (**D**) LGBM, (**E**) LR, (**F**) RF, and (**G**) SVM.

**Figure 3 ijms-25-07049-f003:**
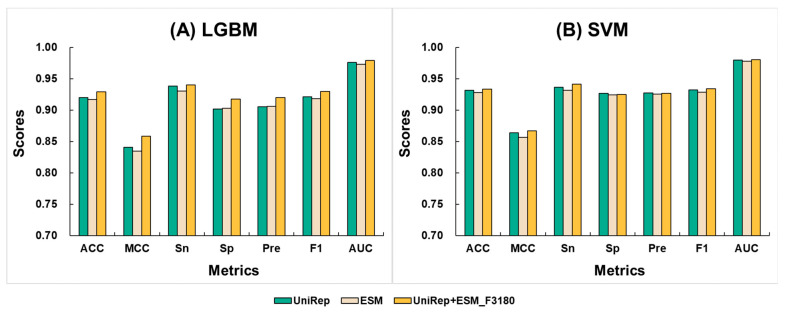
Comparison of the average values from 10-fold cross-validation and an independent test between fused feature models and single feature models. Here, UniRep is represented in green, ESM in light yellow, and UniRep+ESM_F3180 in dark yellow. The machine learning algorithms include (**A**) LGBM (parameters: {‘num_trees’: 1300, ‘learning_rate’: 0.28}) and (**B**) SVM (parameters: {‘C’: 1.9306977288832496, ‘gamma’: ‘scale’}).

**Figure 4 ijms-25-07049-f004:**
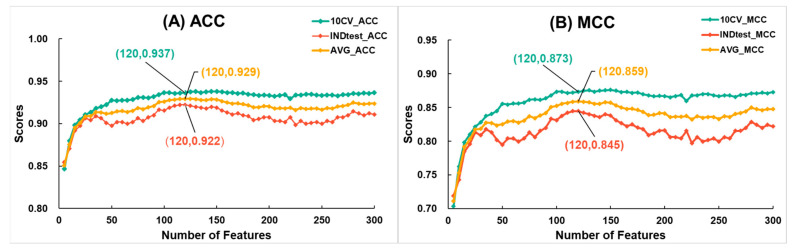
The variation in (**A**) accuracy and (**B**) Matthews correlation coefficient during the feature selection process for ESM+UniRep_F3180 with the number of features. Here, 10-fold cross-validation metrics are represented in green, independent test metrics in red, and the average of cross-validation and independent test metrics in yellow. LGBM Classifier parameters: {‘num_leaves’: 32, ‘n_estimators’: 888, ‘max_depth’: 12, ‘learning_rate’: 0.16, ‘min_child_samples’: 50, ‘random_state’: 2020, ‘n_jobs’: 8}.

**Figure 5 ijms-25-07049-f005:**
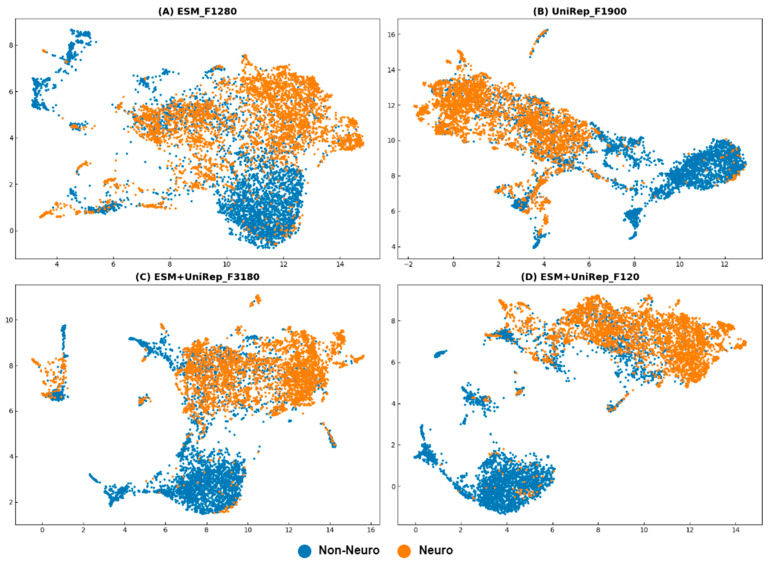
UMAP visualization plots (parameters: {‘metric’: ‘correlation’, ‘n_neighbors’: 45, ‘min_dist’: 0.12}). (**A**) using the ESM training set; (**B**) using the UniRep training set; (**C**) using the ESM+UniRep_F3180 training set; and (**D**) using the ESM+UniRep_F120 training set. Neuropeptides are represented by orange dots, and non-neuropeptides are represented by blue dots.

**Table 1 ijms-25-07049-t001:** Comparison of 10-fold cross-validation accuracy for six encoding methods across seven machine learning algorithms.

Metrics	10-Fold Cross-Validation_ACC						
Features/Algorithms	GNB ^1^	KNN ^2^	LDA ^3^	LGBM ^4^	LR ^5^	RF ^6^	SVM ^7^
UniRep	0.729	** 0.895 **	0.894	0.918	0.906	** 0.908 **	0.920
ESM	** 0.764 ^8^ **	0.889	** 0.906 **	** 0.920 **	** 0.916 **	0.901	** 0.924 **
SSA	0.615	0.826	0.822	0.881	0.826	0.858	0.871
LM	0.757	0.840	0.876	0.909	0.872	0.890	0.910
BiLSTM	0.733	0.850	0.848	0.908	0.885	0.888	0.917
TAPE_BERT	0.751	0.871	0.888	0.890	0.878	0.874	0.912

^1^ GNB parameter: {‘default’}, ^2^ KNN parameters: {‘n_neighbors’: 5}, ^3^ LDA parameters: {‘default’}, ^4^ LGBM parameters: {‘n_estimators’: 500}, ^5^ LR parameters: {‘penalty’: ‘l2’, ‘C’: 10, ‘max_iter’: 1000, ‘solver’: ‘sag’}, ^6^ RF parameters: {‘n_estimators’: 500}, ^7^ SVM parameters: {‘kernel’: ‘rbf’, ‘gamma’: ‘auto’, ‘C’: 1, ‘probability’: ‘True’}, ^8^ the best value of each column is underline and in bold.

**Table 2 ijms-25-07049-t002:** Comparison of 10-fold cross-validation and independent test metrics for SVM-based models after feature fusion and feature selection.

Algorithm = SVM	10-Fold Cross-Validation							Independent Test						
Features/Metrics	ACC	MCC	Sn	Sp	Pre	F1	AUC	ACC	MCC	Sn	Sp	Pre	F1	AUC
UniRep ^1^	0.932	0.864	0.937	0.927	0.928	0.932	0.980	0.920	0.840	0.917	0.923	0.923	0.920	0.975
ESM ^2^	0.928	0.857	0.932	0.924	0.925	0.929	0.978	0.919	0.838	0.912	0.926	0.925	0.918	0.975
UniRep+ESM_F3180 ^3^	0.933	0.867	** 0.942 **	0.925	0.927	** 0.934 **	** 0.981 **	** 0.928 **	** 0.856 **	** 0.930 **	0.926	0.926	** 0.928 **	** 0.979 **
UniRep+ESM_F120 ^4^	** 0.937 ^5^ **	** 0.873 **	0.940	** 0.933 **	** 0.980 **	** 0.934 **	0.937	0.922	0.845	0.917	** 0.928 **	** 0.966 **	0.927	0.922

^1^ SVM parameters: {‘C’: 17.433288221999874, ‘gamma’: 0.0012689610031679222, ‘kernel’: ‘rbf’}, ^2^ SVM parameters: {‘C’: 14.677992676220706, ‘gamma’: 0.001}, ^3^ SVM parameters: {‘C’: 1.9306977288832496, ‘gamma’: ‘scale’}, ^4^ SVM parameters: {‘C’: 1.873817422860383, ‘gamma’: 0.019306977288832496, ‘kernel’: ‘rbf’}, ^5^ the best value of each column is underline and in bold.

**Table 3 ijms-25-07049-t003:** Comparison of independent test metrics between iNP_ESM and existing methods.

Methods	ACC	MCC	Rec	Pre	F1	AUC
iNP_ESM_F3180	** 0.928 ^1^ **	** 0.856 **	0.930	0.926	** 0.928 **	** 0.981 **
iNP_ESM_F120	0.922	0.845	0.917	** 0.966 **	0.927	0.937
PredNeuroP	0.864	0.738	0.782	0.935	0.852	-
NeuroPred-FRL	0.861	0.740	0.757	0.960	0.847	-
NeuroPpred-Fuse	0.905	0.813	0.908	0.906	0.907	-
NeuroPred-PLM	0.922	0.845	** 0.941 **	0.907	0.924	-

^1^ The best value of each column is underline and in bold.

## Data Availability

The raw data supporting the conclusions of this article will be made available by the authors on request.

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
