# Peer review of "iNP_ESM: Neuropeptide Identification Based on Evolutionary Scale Modeling and Unified Representation Embedding Features"

_ijms, 2024, doi:10.3390/ijms25137049_

Round 1

Reviewer 1 Report

Comments and Suggestions for Authors

The objectives of the study are clearly stated. However, they could be improved by providing more specific details about the expected outcomes and potential applications of the research.

Rows 331-332: Do not insert entire links in the body of the text. Cite it as a reference.

Provide more specific details about the NeuroPep 2.0 dataset, such as the number of neuropeptides included, their distribution, and the criteria for their inclusion or exclusion.

Explain why a threshold value of 0.9 was chosen in CD-HIT to remove sequences exhibiting over 90% similarity to other sequences.

Provide more details about the five encoding methods used to represent neuropeptides for comparison. how were these methods implemented? What parameters were used? Why were these specific methods chosen?

Provide more details about the seven machine learning algorithms used. what parameters were used in each algorithm? Why were these specific algorithms chosen?

Provide more details about how LGBM was used for feature selection. what parameters were used? Why was LGBM chosen for feature selection?

Provide more details about how UMAP was used for dimensionality reduction and visualization. what parameters were used? Why was UMAP chosen for visualization?

Provide more details about the seven performance evaluation metrics adopted. how were these metrics calculated? Why were these specific metrics chosen?

Author Response

Manuscript ID: ijms-3051030

Title: iNP_ESM: Neuropeptides Identification Based on Evolutionary Scale Modeling and Unified Representation Embedding Features

Authors: Honghao Li, Liangzhen Jiang, Kaixiang Yang, Shulin Shang, Mingxin Li, Zhibin Lv *

Response to Reviewer 1 Comments

Dear reviewer,

Thank you for your valuable and positive feedback on our manuscript ijms-3051030. Your constructive comments have been instrumental in enhancing the quality of our paper. As per your suggestions, we have carefully revised the manuscript. Our point-by-point responses are summarized below. For clarity, the reviewers' comments are labeled as C (comment), and our responses are labeled as R (response). The main revisions in our manuscript are highlighted in red within the main text, which can be seen in the MS Word "Track Changes" mode under "No Markup" status. A detailed record of changes can be found in the MS Word "Track Changes" mode under "All Markup" status. Here, we address the reviewers' comments.

C1. Rows 331-332: Do not insert entire links in the body of the text. Cite it as a reference.

R1. Thanks for your good comments. We have removed the relevant hyperlinks in Section 4.2 and cited them as references [23] and [44].

C2. Provide more specific details about the NeuroPep 2.0 dataset, such as the number of neuropeptides included, their distribution, and the criteria for their inclusion or exclusion.

R2. Thank you very much for your suggestion. We have added more details about NeuroPep 2.0 in Section 4.1: NeuroPep 2.0 contains 11,417 unique neuropeptide records, nearly double that of its initial version. Neuropeptides included in the database must be verified through scientific literature or database annotations. These records are derived from 544 articles and 3,688 precursor proteins from the UniProtKB database, covering 924 species and distributed across 81 neuropeptide families.

C3. Explain why a threshold value of 0.9 was chosen in CD-HIT to remove sequences exhibiting over 90% similarity to other sequences.

R3. Thank you for your valuable feedback. By removing highly similar sequences, we can ensure that each sequence in the dataset is unique, thereby avoiding model overfitting. We have added an explanation in Section 4.1: Subsequently, to avoid model overfitting, the remaining dataset was processed using CD-HIT with a 0.9 threshold to remove sequences with more than 90% similarity to other sequences. Following these filtration processes, a total of 4,463 neuropeptides were retained.

C4. Provide more details about the five encoding methods used to represent neuropeptides for comparison. how were these methods implemented? What parameters were used? Why were these specific methods chosen?

R4. Thanks for your kind suggestion. The reason we chose to compare and select these five encoding methods is that they performed well in previous polypeptide sequence-related studies conducted in our laboratory (see reference [29]). For the features that performed well in this project, ESM and UniRep, we provided a detailed explanation of their principles. For the other features, we have also added introductions in Sections 4.2.3 and 4.2.4. Additionally, we have added information on how to obtain the model parameters in Section 4.2.

C5. Provide more details about the seven machine learning algorithms used. what parameters were used in each algorithm? Why were these specific algorithms chosen?

R5. Thank you for your valuable suggestions. The reason we chose these machine learning algorithms is that they are commonly used classical machine learning classification algorithms, which have achieved significant results in the field of bioinformatics (see references in Section 4.3). In addition, we have updated the table footers and figure captions to show the parameters of machine learning algorithms used in the paper and supplementary materials.

C6. Provide more details about how LGBM was used for feature selection. what parameters were used? Why was LGBM chosen for feature selection?

R6. Thank you for your suggestion. We have supplemented the parameters of the feature selection algorithms in the Figure 4 caption. We have also added an explanation in Section 4.4: In previous studies on polypeptide sequences conducted in our laboratory [29], we tested various feature selection methods, including LGBM, ANOVA, and Chi-Squared. Ultimately, we chose the LGBM model due to its efficiency, simplicity, and ability to provide clear ranking results [63].

C7. Provide more details about how UMAP was used for dimensionality reduction and visualization. what parameters were used? Why was UMAP chosen for visualization?

R7. Thank you for your suggestion. The main reason for choosing UMAP for visualization is its higher processing speed when handling high-dimensional data, as we explained in Section 4.5. Additionally, we have included an introduction to several important parameters used for UMAP in Figure 5.

C8. Provide more details about the seven performance evaluation metrics adopted. how were these metrics calculated? Why were these specific metrics chosen?

R8. Thank you for your suggestion. The calculation methods for the seven machine learning evaluation metrics used are presented in Section 4.6. We chose these metrics because they are widely used in binary classification tasks in machine learning (see references [10-16]). Additionally, to ensure the comprehensiveness of the data, we have also updated the tables and figures with another important metric, the Area Under the Curve (AUC).

Finally, we would like to express our sincere gratitude for taking the time out of your busy schedule to review and edit our paper. We are deeply appreciative of your constructive feedback, which has significantly enhanced the quality and presentation of our manuscript.

Sincerely,

Zhibin Lv

Email: lvzhibin@pku.edu.cn

Reviewer 2 Report

Comments and Suggestions for Authors

I appreciate the authors for presenting this clinically useful research article emphasizing the identification of neuropeptides based on evolutionary scale modeling and unified representation embedding features. This is a well-organized research article. The results are robust and support the hypothesis. My comments are as follows:

1. In Tables 1, 2, and 3, why was the area under the curve (AUC) not used to compare these parameters?

2. In Table 3, how can we determine that iNP_ESM is significantly better than other existing methods?

Author Response

Manuscript ID: ijms-3051030

Title: iNP_ESM: Neuropeptides Identification Based on Evolutionary Scale Modeling and Unified Representation Embedding Features

Authors: Honghao Li, Liangzhen Jiang, Kaixiang Yang, Shulin Shang, Mingxin Li, Zhibin Lv *

Response to Reviewer 2 Comments

Dear reviewer,

Thank you for your valuable and positive feedback on our manuscript ijms-3051030. Your constructive comments have been instrumental in enhancing the quality of our paper. As per your suggestions, we have carefully revised the manuscript. Our point-by-point responses are summarized below. For clarity, the reviewers' comments are labeled as C (comment), and our responses are labeled as R (response). The main revisions in our manuscript are highlighted in red within the main text, which can be seen in the MS Word "Track Changes" mode under "No Markup" status. A detailed record of changes can be found in the MS Word "Track Changes" mode under "All Markup" status. Here, we address the reviewers' comments.

C1. In Tables 1, 2, and 3, why was the area under the curve (AUC) not used to compare these parameters?

R1. We greatly appreciate this excellent suggestion. We have revised Figures 2 and 3, as well as Tables 2 and 3 and the supplementary materials, to include experimental data on AUC. Additionally, we have added a comparison of AUC in the discussion section. Initially, we did not use AUC because prior methods did not systematically compare AUC in their literature, resulting in a lack of relevant data.

C2. In Table 3, how can we determine that iNP_ESM is significantly better than other existing methods?

R2. Thank you for your comments. According to Table 3, our iNP_ESM_F3180 significantly outperforms existing methods in both ACC (0.928) and MCC (0.856). Additionally, although iNP_ESM_F3180's Recall and Precision are not the highest, their harmonic mean, the F1 score, still achieves the highest value among all methods (0.928). Therefore, we believe that iNP_ESM outperforms other existing methods.

Finally, we would like to express our sincere gratitude for taking the time out of your busy schedule to review and edit our paper. We are deeply appreciative of your constructive feedback, which has significantly enhanced the quality and presentation of our manuscript.

Sincerely,

Zhibin Lv

Email: lvzhibin@pku.edu.cn

Round 2

Reviewer 2 Report

Comments and Suggestions for Authors

The revised manuscript has replied my comments  point-by-point. Accept is my final decision